# Molecular modelling of the HCMV IL-10 protein isoforms and analysis of their interaction with the human IL-10 receptor

Simone Queiroz Pantaleão[1] , Lívia de Moraes Bomediano Camillo[2] , Tainan Cerqueira Neves[2], Isabela de Godoy Menezes[2], Lucas Matheus Stangherlin[2], Helena Beatriz de Carvalho Ruthner Batista[3], Emma Poole[4], Michael Nevels [5], Eric Alisson Philot[1], Ana Ligia Scott[1], Maria Cristina Carlan da Silva [2]*

1 Center for Mathematics, Computing and Cognition, Federal University of ABC, Santo André, Brazil,
2 Center for Natural and Humanities Sciences, Federal University of ABC, São Bernardo do Campo, Brazil,
3 Pasteur Institute, São Paulo, Brazil, 4 Department of Medicine, University of Cambridge, Cambridge, United Kingdom, 5 School of Biology, University of St Andrews, St Andrews, United Kingdom

☯ These authors contributed equally to this work.
* cristina.carlan@ufabc.edu.br

**Data Availability Statement:** All used files are available from a Kaggle database (https://www.

## Abstract

The human cytomegalovirus (HCMV) UL111A gene encodes several homologs of the cellular interleukin 10 (cIL-10). Alternative splicing in the UL111A region produces two relatively well-characterized transcripts designated cmvIL-10 (isoform A) and LAcmvIL-10 (isoform B). The cmvIL-10 protein is the best characterized, both structurally and functionally, and has many immunosuppressive activities similar to cIL-10, while LAcmvIL-10 has more restricted biological activities. Alternative splicing also results in five less studied UL111A transcripts encoding additional proteins homologous to cIL-10 (isoforms C to G). These transcripts were identified during productive HCMV infection of MRC-5 cells with the high passage laboratory adapted AD169 strain, and the structure and properties of the corresponding proteins are largely unknown. Moreover, it is unclear whether these protein isoforms are able to bind the cellular IL-10 receptor and induce signalling. In the present study, we investigated the expression spectrum of UL111A transcripts in fully permissive MRC-5 cells and semi permissive U251 cells infected with the low passage HCMV strain TB40E. We identified a new spliced transcript (H) expressed during productive infection. Using computational methods, we carried out molecular modelling studies on the three-dimensional structures of the HCMV IL-10 proteins encoded by the transcripts detected in our work (cmvIL-10 (A), LAcmvIL-10 (B), E, F and H) and on their interaction with the human IL-10 receptor (IL-10R1). The modelling predicts clear differences between the isoform structures. Furthermore, the *in silico* simulations (molecular dynamics simulation and normal-mode analyses) allowed us to evaluate regions that contain potential receptor binding sites in each isoform. The analyses demonstrate that the complexes between the isoforms and IL-10R1 present different types of molecular interactions and consequently different affinities and stabilities. The knowledge about structure and expression of specific viral IL-10 isoforms has implications for understanding of their properties and role in HCMV immune evasion and pathogenesis.

kaggle.com/datasets/tainanneves/data-availability-hcmv-il10-molecular-models).

**Funding:** Funder 01: Grant number: 2020/08527-1 Fundação de Amparo a Pesquisa do Estado de São Paulo, Brazil (FAPESP) - https://fapesp.br/ Author: Maria Cristina Carlan da Silva - Maria Cristina Carlan Silva Funder 02: Grant number: 2020/07767-9 Fundação de Amparo a Pesquisa do Estado de São Paulo, Brazil (FAPESP) - https://fapesp.br/ Author: Tainan Cerqueira Neves - Neves, Tainan C. Funder 03: Grant number: 164052/2020-8 Conselho Nacional de Desenvolvimento Científico e Tecnológico (CNPq) - https://www.gov.br/cnpq/pt-br Author: Simone Queiroz Pantaleão - Pantaleão, Simone Queiroz The funders had no role in study design, data collection and analysis, decision to publish, or preparation of the manuscript.

**Competing interests:** The authors have declared that no competing interests exist.

**Abbreviations:** HCMV, Human cytomegalovirus; cmvIL-10, HCMV interleukin 10 homolog; cIL-10, cellular IL-10; GBM, glioblastoma multiforme; hpi, hours post infection; MOI, multiplicity of infection; nested PCR, nested polymerase chain reaction.

# Introduction

Human cytomegalovirus (HCMV) is a ubiquitous beta herpesvirus in the human population [1] and a life-threatening agent in immunocompromised individuals [2,3]. The virus is also the leading viral congenital infection, causing deafness and central nervous system diseases such as cognitive impairment in neonates [4]. In immunocompetent individuals, primary infection is controlled by the immune system and is generally subclinical. Immune control of viral replication results in elimination of most productively infected cells, but the virus is never fully cleared and stays for a lifetime in the host. HCMV may persist in a state with low levels of replication and ultimately establishes a latent infection, where the viral genome is present in certain cell types, viral expression is limited and no infectious virus is produced [5].

The ability of HCMV to counteract the immune system is imperative for successful infection, and an important protein in this process is the viral homolog of the cellular interleukin-10 (cIL-10), encoded by UL111A [6]. The UL111A gene is known to be differentially spliced and produces cmvIL-10 (175 amino acids) and latency-associated cmvIL-10 (LAcmvIL-10) (139 amino acids) proteins [6,7].

Expression of cmvIL-10 was demonstrated during lytic HCMV infection in fibroblasts. The protein has only 27% sequence identity with cIL-10, but its conformation preserves the ability to bind the IL-10R1 subunit of the cIL-10 receptor with identical affinity [8], leading to signalling with several immunomodulatory outcomes similar to cIL-10 [9–13].

LAcmvIL-10 is expressed during lytic and latent infection [7,14]. In contrast to cmvIL-10, it has restricted immunomodulatory properties, such as an ability to inhibit transcription of components of the MHC class II biosynthesis pathway and production of MHC class II proteins in latently infected granulocyte-monocyte progenitor cells [14]. Five other transcripts and their encoded proteins, resulting from alternative splicing in the UL111A region, have been reported to be expressed in productively infected MRC-5 cells with the HCMV laboratory strain AD169. To facilitate the nomenclature, these transcripts were designated C, D, E, F and G (cmvIL-10 and LAcmvIL-10 were referred to as A and B, respectively) [15].

Although several studies reported functional properties of cmvIL-10 (A) and LAcmvIL-10 (B), their expression in cells infected with a low passage HCMV strain is unknown. Likewise, there is little if any information about the structure, ability to bind the cIL-10 receptor and functions of transcripts C to G. Here, we evaluated the expression of UL111A transcripts in MRC-5 and U251 cells infected with the HCMV TB40E strain and detected by nested PCR transcripts A, B, E and F in addition to a newly identified transcript termed H.

Using *in silico* analysis which includes molecular modelling, identification and characterization of binding sites, normal modes analysis as well as molecular dynamics and docking, we demonstrate structural differences between these HCMV IL-10 isoforms and indicate potential residues that could establish an interaction with the IL-10R1 subunit of the cIL-10 receptor. We also demonstrate that the complexes between the different isoforms and IL-10R1 have distinct molecular interactions and consequently different stabilities.

# Materials and methods

## Cells and virus

MRC-5 human embryonic lung fibroblast cells were obtained from the American Type Culture Collection (ATCC) and human glioblastoma multiforme (GBM) cell line U251 was provided by Dr. Eugenia Costanzi-Strauss, University of São Paulo. The cells were maintained in Dulbecco's Modified Eagle's Medium (DMEM) supplemented with 10% fetal bovine serum,

100 U/ml penicillin and 100 μg/ml streptomycin (Thermo Fisher Scientific) at 37˚C in a humidified atmosphere with 5% $CO_2$.

The low-passage HCMV strain TB40E used in this study has been described [16]. For production of virus, TB40E bacterial artificial chromosome (BAC) DNA was used to transfect MRC-5 cells by electroporation. When all cells displayed a cytopathic effect, supernatants were collected and infectious yields were determined by plaque assay on MRC-5 cells.

### Infections and PCR

MRC-5 cells were plated in 12 well plates ($10^6$ cells/well) and exposed to the HCMV TB40E strain at a multiplicity of infection (MOI) of 1. Total RNA was extracted from the cells and reverse transcribed using the ImProm-II kit (Promega) according to the manufacturer's instructions. The cDNAs were subjected to nested PCR for amplification of the UL111A sequences. The outer and inner primers were designed using Primer3 Plus (http://www. bioinformatics.nl/cgi-bin/primer3plus/primer3plus.cgi). Outer primers: forward primer 5'- ATG CTG TCG GTG ATG GTC TC−3' and reverse primer 5'-CTA CTT TCT CGA GTG CAG ATA CTC T-3'. Inner primers: forward primer 5'-TCT AGG CGC TTC CGA GGA G- 3' and reverse primer 5'-TCC AAC TCG CTG AGA CCT TTC-3'.

For both rounds of nested PCR, 1 U of Taq DNA polymerase, 10x PCR buffer (200 mM Tris-HCl pH 8.4, 500 mM KCl), 1.5 mM $MgCl_2$ and 0.2 mM dNTPs were used in a final volume of 25 μL per reaction. In the first PCR, 0.2 mM of outer primers and 1 μL of cDNA were used, whereas in the second reaction, 0.2 mM of inner primers and 1 μL of the first reaction were employed. A T100 thermal cycler (Bio-Rad) was used for all reactions. PCR conditions were the same for both reactions, beginning with DNA denaturation at 95˚C for 5 min and 30 amplification cycles (95˚C for 1 min, 61˚C for 1 min and 72˚C for 1 min) followed by a final extension of 5 min at 72˚C. Nested PCR products were observed after electrophoresis in an 8% non-denaturing acrylamide gel stained with GelRed (Sigma-Aldrich). The bands observed at each time point of all infected cells were purified using the GenElute Gel Extraction kit (Sigma-Aldrich).

### Sequence analysis

The DNA products obtained by nested PCR were sequenced using the Dye Terminator Cycle Sequencing Kit with AmpliTaq DNA polymerase in an ABI 3500 Genetic Analyzer (Applied Biosystems).

Sequence analyses were performed with BioEdit software. The obtained sequences were aligned to the published human IL-10 (cIL-10) (Gene ID 3586) and TB40E cmvIL-10 (Uniprot ID Q6SWGO), indicating high levels of homology. As a reference, we used the UL111A transcripts expressed by the human herpesvirus 5 strain TB40E clone Lisa, complete genome (GenBank: KF297339.1). Sequence alignments and similarities were calculated with Clustal Omega 1.2.2 [17]. Our new sequence data of UL111A transcripts can be accessed through NCBI Gene-Bank accession numbers: OP346856, OP346857, OP346858, OP346859, OP346860 and OP346861.

### Molecular modelling, evaluation and selection

The three-dimensional structural models of the isoforms A (cmvIL-10), B (LAcmvIL-10), E, F and H were generated in MODELLER v.10 [18] using the structure of cIL-10 solved experimentally by x-ray diffraction (PDB: 1Y6K) [8] as a template model. A model of isoform A (cmv IL-10) was generated in addition to the structure solved experimentally by x-ray diffraction (PDB: 1LQS) [8].

The final selected models were submitted to a molecular dynamic simulation of 100 ns, performed using NAMD v.2.14 [19], to obtain energy relaxed conformation in solution. The solution input for molecular dynamics was generated by Charmm-Gui [20], using an octahedral box with the solvated protein and KCl ions included by Monte-Carlo method [21] for neutralization.

In order to rule out structures with distorted bond angles, all models were evaluated using ProSA webserver [22] and Verify 3D [23]. MolProbity v.4.4 [24] was used to provide Ramachandran plots and geometry statistics of the structures (S1 Fig).

### Search and characterization of binding sites

Complementary to the above analyses, we performed a search and characterization of regions with binding site properties for each isoform. PeptiMap [25,26] was used to detect regions suitable for peptide binding and FTMap [27–29] was used for characterization of the physicochemical profiles of these regions. The modelled structures of isoforms A, B, E, F, H and IL-10R1 were submitted to the PeptiMap server and the contribution of each amino acid residue of the protein to establish hydrogen bonding type contacts and unbonded contacts was verified, validating the regions indicated as probable binding sites (site characterization step). After that, the volume and area of these binding sites was calculated using Chimera 1.14 [30] (S1 Table).

### Molecular docking

To obtain atomic-level structures of the complex between the protein isoforms and IL-10R1 (PDB ID: 1LQS_1), starting from their unbound components, we performed CPORT [31] and HADDOCK 2.4 [32,33] analysis. CPORT was used to predict charged positive and negative residues in the interface of the viral IL-10/IL-10R1 complex. The residue information was used for flexible molecular docking using HADDOCK 2.4, and several poses were generated. The poses were clustered, and to obtain energy relaxed conformation in solution the most populated and highest scoring poses of each isoform were selected.

The interactions between protein and receptor were analysed using PIC Server [34], LIGPLOT [35] and BINANA [36] in order to provide information on specific interactions established between the complex components. This approach helped to identify those complexes that are remarkable for selected binding features, in addition to screening for lead candidates with specific and desirable binding characteristics.

### Normal modes and molecular dynamics

The DynOmics web server [37] was used to determine the structural dynamics of the complexes between the various isoforms (A, B, E, F and H) and IL-10R1. This technique is based on elastic network models to obtain energy relaxed conformation in solution. For the complex that presented the best binding conditions, the molecular dynamics technique was performed, in this case isoform F and IL-10R1. Molecular dynamics was performed with GROMACS [38] for the most promising complex to evaluate stability through several minimization steps to achieve the lowest energy complex. The solution input for molecular dynamics was generated by [20] using an octahedral box with the complex solved and KCl ions included by Monte-Carlo method [21] for neutralisation.

## Results

### Expression of HCMV IL-10 transcripts in permissive cells

It has been previously demonstrated that MRC-5 cells infected with the high passage, laboratory HCMV strain AD169 express several alternative viral IL-10 mRNAs, in addition to the

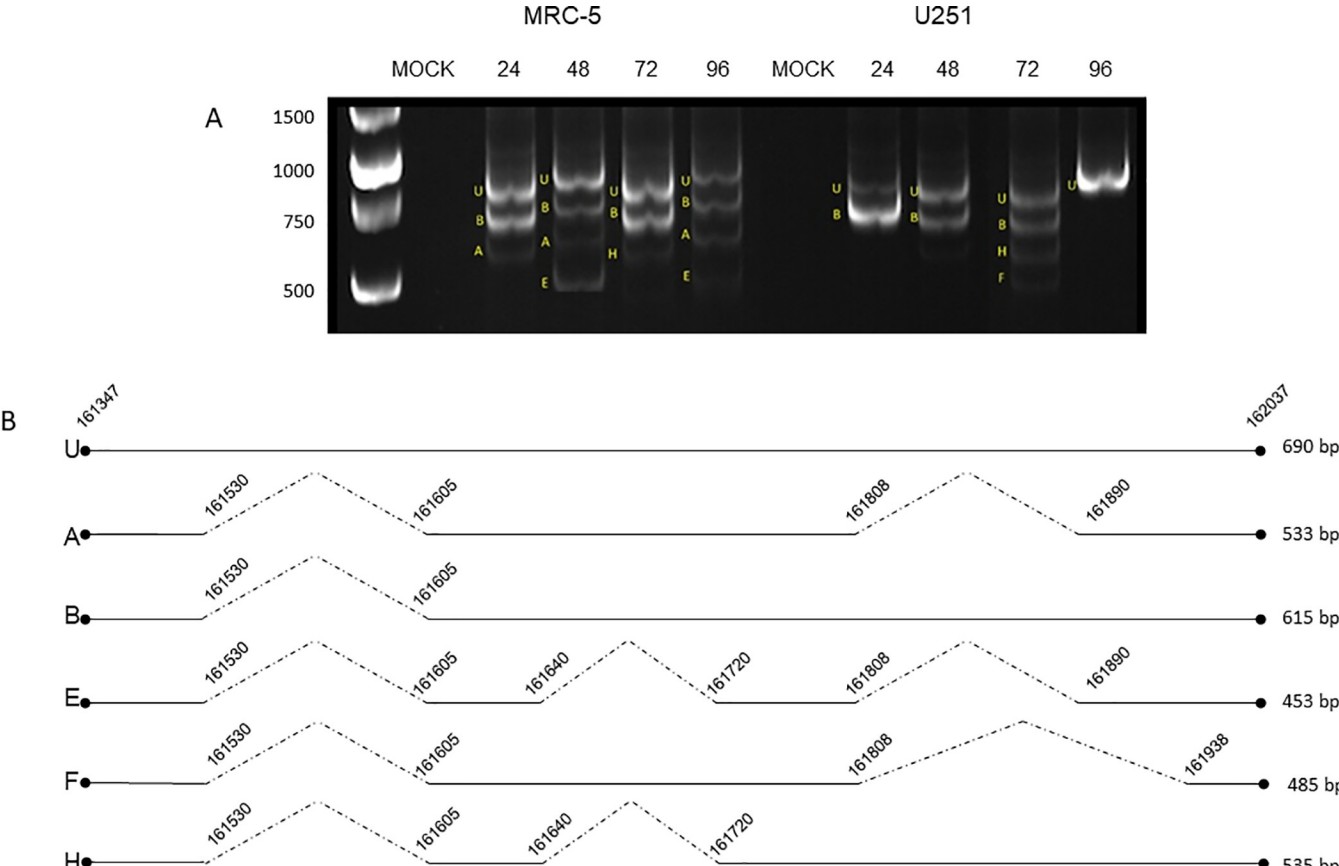

**Fig 1. HCMV IL-10 transcripts detected in infected cells.** (A) Acrylamide gel electrophoresis showing the HCMV IL-10 transcripts obtained by nested PCR, in MRC-5 and U251 cells, infected for the time points indicated. The identity of the transcripts determined by DNA sequencing is indicated by the letters (U) unspliced; (A) transcript A (cmvIL-10); (B) transcript B (LAcmvIL-10); transcript E, F and H. (B) Squematic illustration of HCMV IL-10 pre-mRNA (unspliced) and the transcripts identified. Exon (continuous lines) and intron (dotted lines) positions are shown according to the HCMV TB40e strain (gene Bank: KF297339). The start and end positions of each intron are numbered, relative to the TB40E strain.

canonical transcripts cmvIL-10 and LAcmvIL-10 [15]. Although used for many decades for in vitro studies, the genomic integrity of high passage HCMV strains is compromised due to accumulations of mutations in culture. Thus, these strains cannot be considered adequate representatives of clinical disease. Nowadays, the development of BACs containing viral genomes with none or few mutations that closely resemble the wild-type clinical isolate has facilitated HCMV studies, especially those related to pathogenesis [39]. Therefore, we investigated the presence of HCMV IL-10 transcripts in MRC-5 and U251 GBM cells, which have different degrees of permissiveness, with the low passage HCMV TB40E strain. These cells were infected over a time course (24 to 96 h) and then analysed by nested PCR (Fig 1). The amplified PCR products were separated by acrylamide gel electrophoresis and subjected to DNA sequencing. Alignment of the identified TB40E sequences with the transcripts produced in cells infected with AD169, as reported by Lin et al (2008), demonstrated complete identity except for one additional codon at amino acid position 30, encoding a threonine [15]. This residue is present in a threonine stretch previously shown to be of variable size in different HCMV strains [40]. Due to the high similarity between strains, we adopted the nomenclature for HCMV IL-10 transcripts proposed by Lin et al (2008) in this study: transcript A (cmvIL-10), transcript B (LAcmvIL-10), transcripts C, D, E, F and G [15].

In MRC-5 cells, in addition to the unspliced (U) mRNA, transcripts A and B were detected at all four time points but 72 h, and transcript E was detected at 48 and 96 h. In U251 cells, transcript B was predominant at 24, 48 and 72 h and was not detected at 96 h, and transcript F was identified at 72 h post infection (Fig 1). Moreover, a transcript that had not been described before was detected in MRC-5 and U251 cells infected for 72 h (Fig 1). Consistent with the existing nomenclature, we are naming this novel mRNA transcript H. Transcripts C, D and G were not detected in infected MRC-5 and U251 cells. A schematic of the HCMV IL-10 transcripts detected in this study is presented in (Fig 1). The first and last nucleotides of each intron are numbered according to the TB40E strain (GenBank: KF297339.1).

**Three-dimensional structure prediction.** All known HCMV IL-10 protein isoforms have the first 60 N-terminal amino acids in common, but they differ in their C-terminal regions. cmvIL-10 binds the cIL-10 receptor and appears to have the same properties as cIL-10, but LAcmvIL-10 has fewer biological properties [14]. The other isoforms (C, D, E, F and G) have been poorly studied. To date, there is no information about the interaction of isoforms B to G with the receptor.

We hypothesized that the HCMV IL-10 isoforms may have different profiles of receptor binding that could lead to different signalling patterns, due to differences in their tertiary structures. To predict the structures of the isoforms encoded by the TB40E transcripts identified in our work, we constructed homology models based on the reported structure of cIL-10 (PDB: 1Y6K) [41].

The generated protein models are displayed in comparison to the structures of cIL-10 and cmvIL-10 (isoform A) (PDB: 1LQS) as reported by Jones et al (2002) (Fig 2). The helixes are named as first described for cIL-10. The structure model of isoform A maintains 99.3% similarity with the structure of cmvIL-10, previously determined by x-ray crystallography, and both lack helix B when compared with cIL-10. Isoform B (LAcmvIL-10), on other hand, lacks helices B, E and F causing a profound alteration in its structure in comparison to cIL-10. Isoform E lacks helixes B and C, and isoform F lacks helix B and E, as compared to cIL-10. Finally, isoform H lacks helices B, C, E and F (Fig 2). Interestingly, the amino acid sequences of isoforms B and H predict an extra helix (Fig 2). However, only isoform B (not isoform H) shows an extra helix in its modelled tertiary structure (Fig 2).

Jones et al (2002) identified amino acids in cmvIL-10 that are likely important for binding to IL-10R1. In the N-terminal portion, these amino acids include ARG27, LYS34 and GLN38, all present in helix A and the AB loop. In the C-terminal portion, the important amino acids are SER141, ASP144 and GLU151 present in helix F [8]. The locations of these residues are shown in the structural models (Fig 2). The variances in amino acid positions between the published structure of cmvIL-10 and the cmvIL-10 (isoform A) modelled in our work (ARG46, LYS58, GLN62, SER159, ASP162, GLU169) result from differences in the viral strain used by our group (TB40E) compared to the strain used by Jones et al (2002) (NCBI: KFKF297339-1) [8]. As expected, the model of isoform A maintains all the residues required for binding to the receptor [8]. Isoforms E and F also maintain all these residues, even though they lack helices C and E, respectively. On the other hand, isoforms B and H do not have the residues SER159, ASP162 and GLU169 due to the lack of helix F in their C-terminal regions (Fig 2).

## Structural dynamics analysis

The three-dimensional models allowed us to verify the presence or absence of previously reported amino acids to be likely required for binding of cmvIL-10 to IL-10R1, in the isoforms detected in this work. In the isoforms lacking these residues, other amino acids may contribute

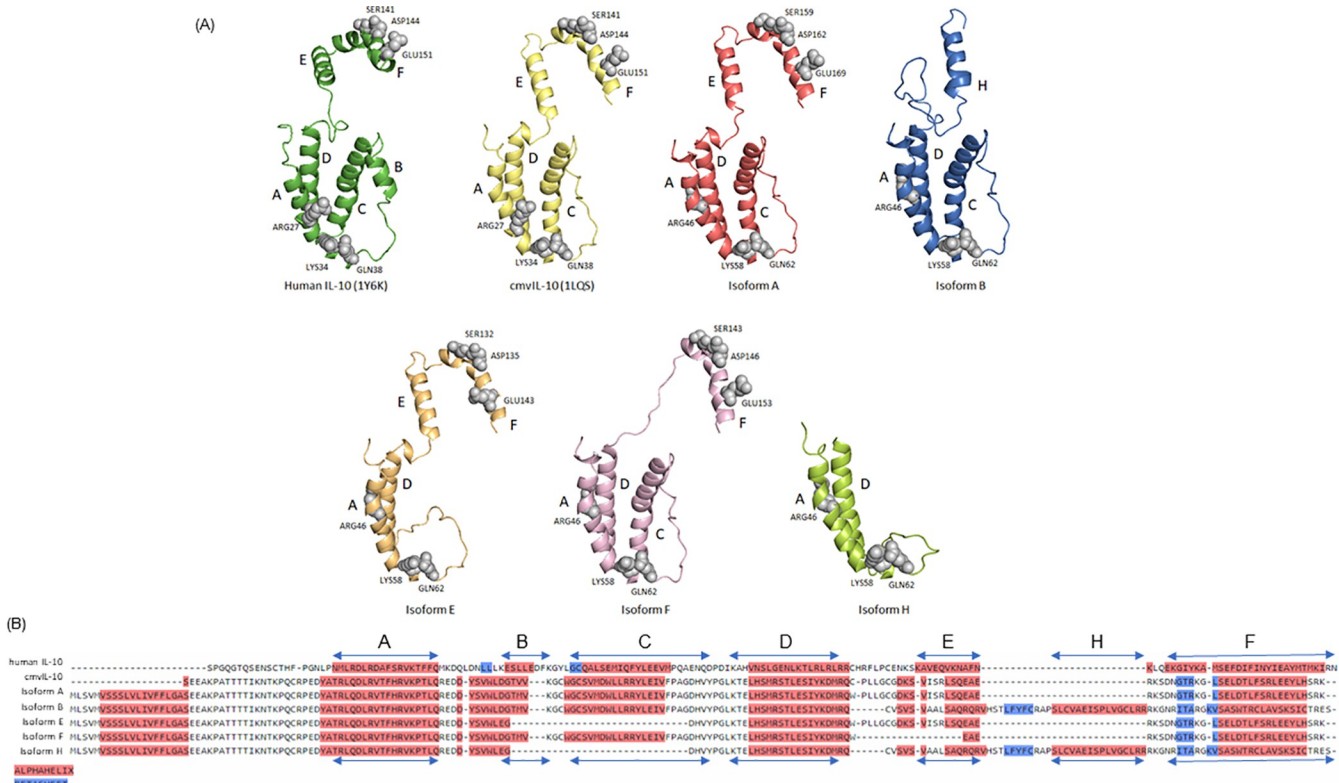

**Fig 2. Structure and amino acid sequences of cIL-10 and HCMV IL-10 isoforms.** (A) Structures of cIL-10 (green) and isoform A (cmvIL-10) (yellow), obtained from Protein Data Bank (PDB) (PDB: 1Y6K), and the modelled structures of isoform A (cmvIL-10 (red), isoform B (LAcmvIL-10) (blue), isoform E (orange), isoform F (pink) and isoform H (light green). The helices A, B, C, D, E, F and H, are highlighted in the structures. Amino acids described as important for cmvIL-10/IL-10R1 binding are shown in grey and identified with the residue name and position. (B) Amino acid alignment between the cIL-10 and HCMV isoforms. Predicted secondary structures are highlighted as red (helix) and blue (beta sheet). The nomenclature adopted for the helix structures (A, B, C, D, E, F and H) are presented.

to receptor binding in a different complex conformation. Therefore, we searched for potential binding sites in each isoform using PeptiMap followed by characterization using FTMap. The results show that all isoforms have up to six binding sites with physicochemical properties (coupling affinity profiles for probe molecules), favoring molecular interactions with small organic molecules, which are standard probe molecules of the tool due to their chemical diversity (acetamide, acetonitrile, acetone, acetaldehyde, methylamine, benzaldehyde, benzene, isobutanol, cyclohexane, N'N-dimethylformamide, dimethyl ether, ethanol, ethane, phenol, isopropanol, and urea). This applies especially to those isoforms containing polar groups, hydrogen bond donors and acceptors, positive and negative charges, or hydrophobic and aromatic moieties. The amino acid residues that make up each binding site in each isoform are described in (S1 Table), and their respective area and volume values can be found in (S1 Table).

The illustrations in Figs 3 and S2–S5 show the binding sites of the isoforms, their surface, the bioactive conformation for docking to IL-10R1 and the representation of the vectors, indicating the direction of natural protein movement. There are stiffer regions with less movement (blue), regions with high movement (pink/red), and regions with intermediate movement (white). Of all potential sites determined in each isoform, the sites considered most likely to contribute to binding are sites 1, 2 and 3 in isoform A; sites 1 and 4 in isoform B; sites 1, 2 and 4 in isoform E; sites 1 and 5 in isoform F; sites 1, 3, 5 and 6 in isoform H and sites 2, 5 and 4 in IL-10R1 (Fig 3) and (S2–S5 Figs).

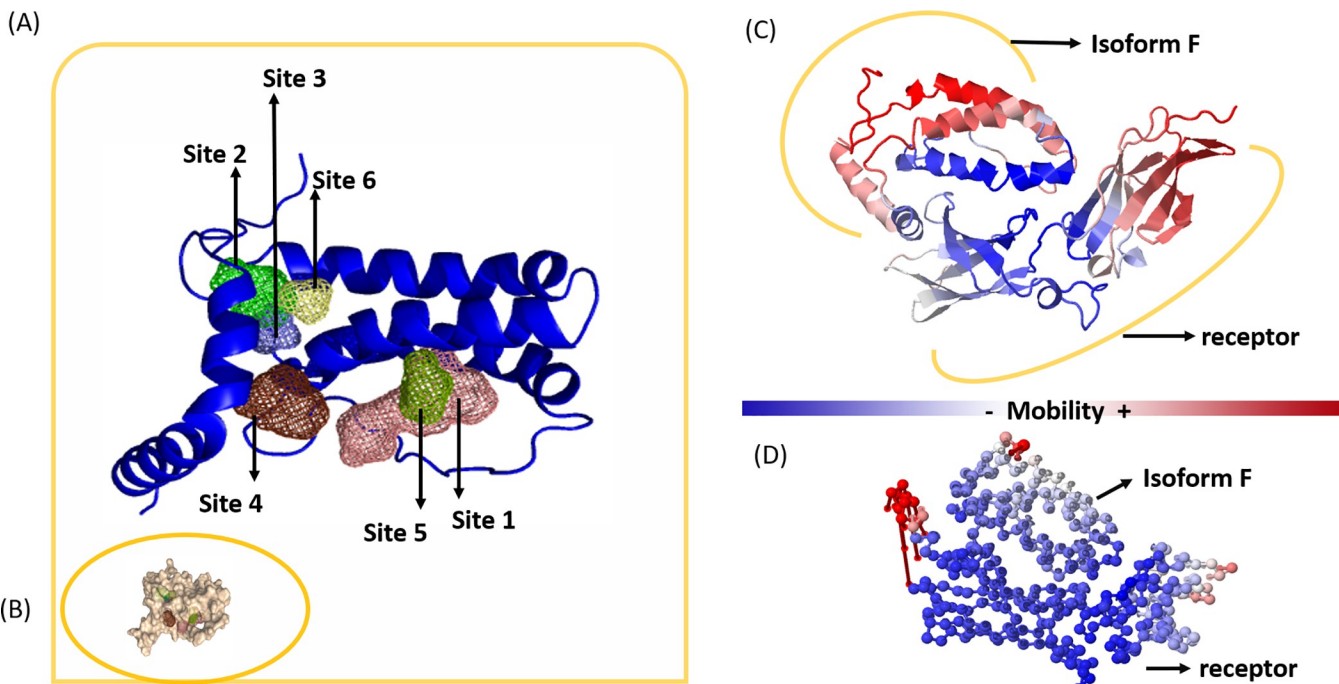

**Fig 3. Particularities of the complex formed between the Isoform F and IL-10R1.** (A) Binding sites detected in the isoforms. Site 1: Pink; site 2: Green; site 3: Lilac; site 4: Brown; site 5: Olive green. (B) Representation of the isoform F on the surface. (C) Bioactive conformation of the isoform F coupled to the IL-10R1. (D) Vectors obtained by calculating the normal modes, where we observe regions with less movement (more rigid) (blue), regions composed of residues with high movement (pink and red), and regions with intermediate movement (white).

The additional sites have high affinity with the probes, indicating good diversity in receiving chemical groups in this region of the protein, such as hydrophobicity, hydrogen bond acceptor and donor groups and aromatic residues. However, they are located in areas of limited access such as loops or buried regions.

## Molecular docking

In the next step, protein-protein docking was performed for all isoforms and IL-10R1. The best poses for each docking were considered, based on several criteria such as binding energy from HADDOCK, interactions for the complex (using BINANA) and the match of binding site localizations with the poses. Complementarity to the data described by Jones et al (2002), we show the possible types of interactions between the receptor and isoforms, including hydrogen bonds, hydrophobic contacts, salt bridges and $\pi$-stacking (S2 Table). We also show the energy of the complexes: isoform A/receptor (-93.1 and $\Delta$G -9.8); isoform B/receptor (-56.9 and $\Delta$G -11.7); isoform E/receptor (-75.3 and $\Delta$G -11.6); isoform F/receptor (-83.1 and $\Delta$G -11.8); isoform H/receptor (-63.4 and $\Delta$G -12.7).

To further understand the possible interactions of the HCMV IL-10 isoforms with the receptor we aimed to evaluate the functional movements in the conformations obtained by the molecular coupling analysis. Specifically, we addressed whether the isoforms were able to inhibit receptor movement when complexed with IL-10R1.

The results indicate that the most rigid complex is isoform F/IL-10R1, possibly due to the greater number of molecular interactions. Besides forming hydrogen-bonding, hydrophobic contacts and salt-bridges, isoform F can also establish T-shaped (Π) interactions with IL-10R1,

which contribute to the observed structural stability. The complex becomes more rigid, suggesting better inhibition (more blue than red residues) or less functional movement of waste.

The regions directly involved in the binding (coupled interfaces) in each complex formed by the isoforms and the receptor could be observed. The key residues in isoform F (present in binding sites 2 to 5) are located in regions of less movement, demonstrating that the coupling is capable of slowing down the movement, thus inhibiting the complex as illustrated in Fig 3. It should also be noted that between residues 80 and 105 (binding sites), isoform F becomes more rigid.

The Root Mean Square Fluctuation (RMSF), which is calculated based on normal modes of the protein giving the flexibility profile, were compared and are shown in S6 Fig. The values of RMSF are the mean value of fluctuation for each $C_\alpha$, calculated for all vibrational normal modes of the protein where the normal modes correspond to the intrinsic global motion of the molecules.

Noteworthy, the RMSF values for the isoform F complex (0.477062) and APO isoform F (0.336129) have a low correlation according to the Pearson test, indicating that when Isoform F binds to IL-10R1, its structural flexibility changes significantly.

The isoform F/IL-10R1 complex is followed, in order of affinity, by the complexes formed by isoform B, isoform E, isoform A or isoform H and IL-10R1 (Figs 3 and S2–S5). The fluctuation of amino acids in the complex formed by isoform F and the receptor is illustrated in (Fig 4).

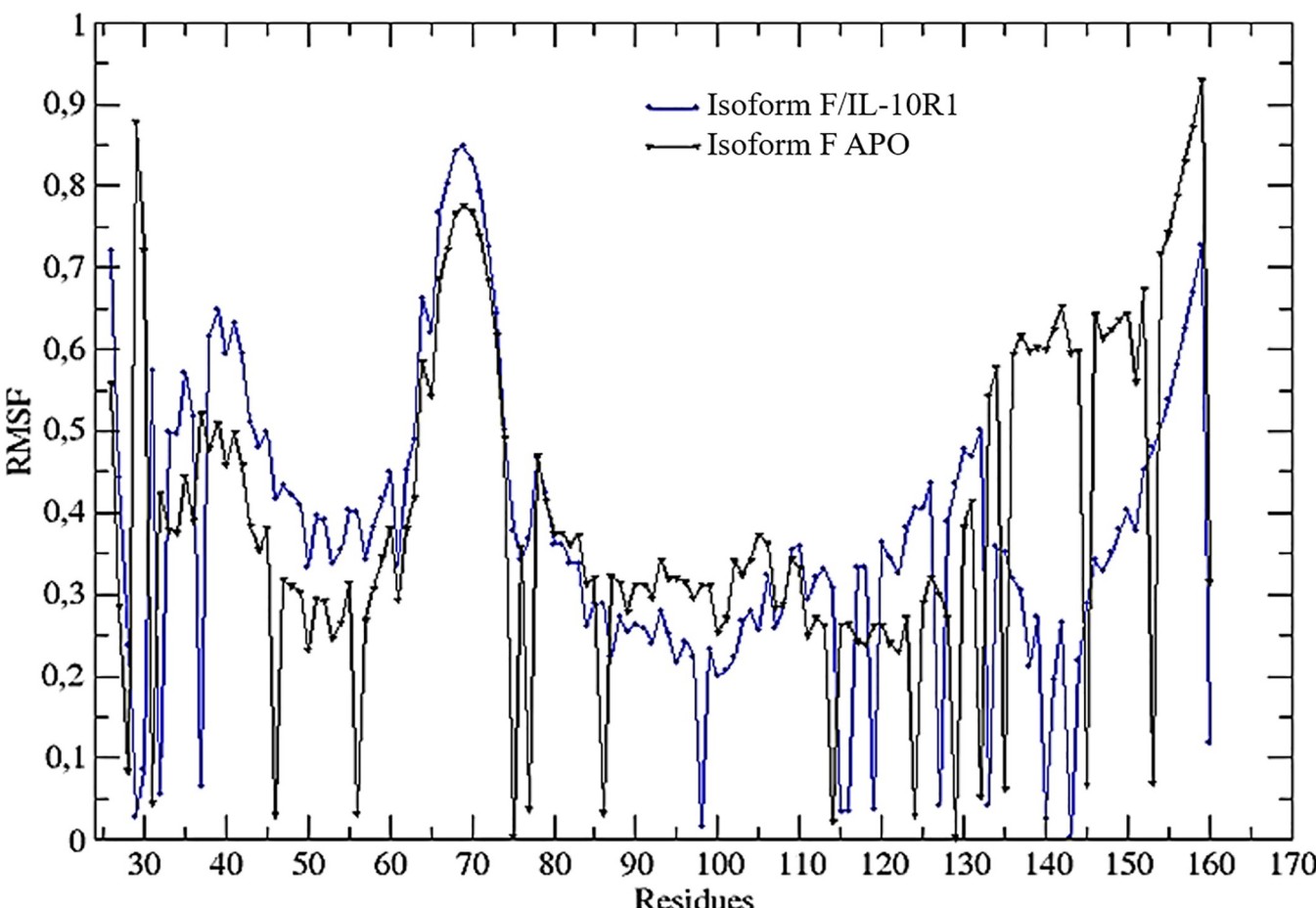

**Fig 4. Difference in root mean square fluctuation (RMSF) Presented values between isoform F complexed with the IL-10R1 and isoform F in Apo conformation, in green and blue respectively.**

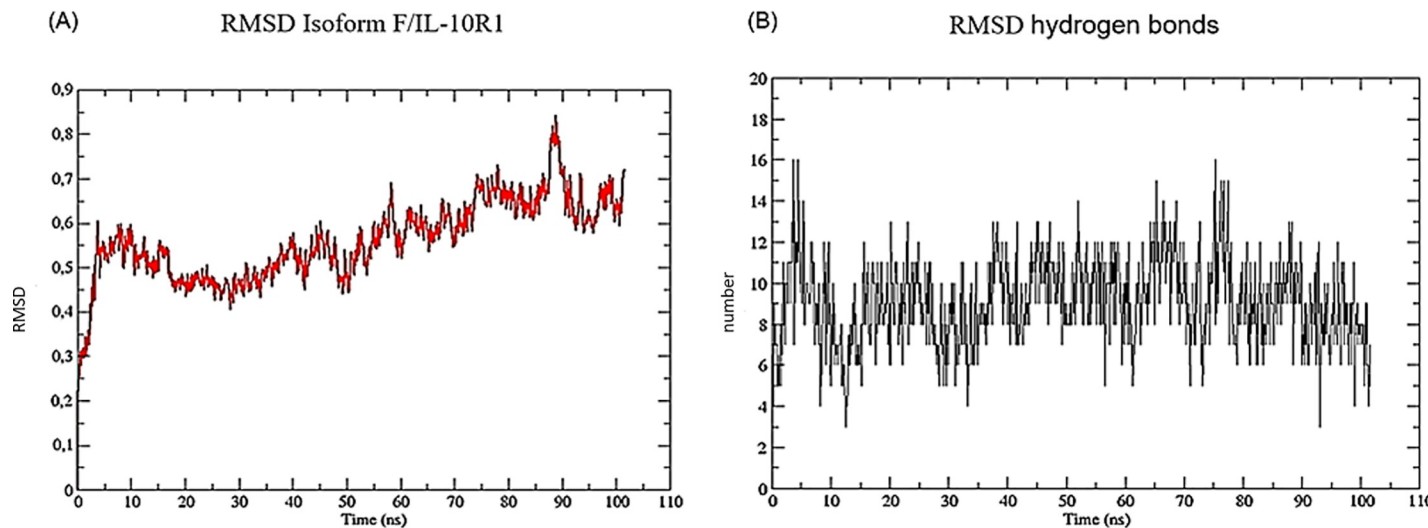

**Fig 5. Stability of the complex formed by isoform F and IL-10R1.** (A) Stability of the complex formed by isoform F and IL-10R1, according to the root mean square fluctuation (RMSD) values. (B) The stability of the hydrogen bonds of the complex formed by isoform F and IL-10R1.

In sum, according to our molecular docking, interaction analysis and ΔG values all isoforms can interact with the receptor. However, a stable complex formation between isoforms F and B with IL-10R1 was observed (Fig 5). These results suggest that the isoforms may have particular interactions, with different affinities and stabilities with IL-10R1 that could result in different functional properties.

## Discussion

The proteins encoded by the HCMV UL111A transcripts, particularly the most extensively studied cmvIL-10 (A) and LAcmvIL-10 (B) isoforms, exert immunosuppressive functions that confer advantages to the virus [9,10,12–14,42,43]. Interestingly, additional transcripts are produced from the UL111A region by alternative splicing and very little is known about the expression, structure, and functions of the encoded proteins.

We show here that MRC-5 and U251 cells infected with the low passage HCMV TB40E strain express cmvIL-10 (A) and LAcmvIL-10 (B), in addition to transcripts E, F and a newly identified transcript denominated H (Fig 1). Three transcripts previously identified by Lin et al (2008) (C, D, and G) were not detected in our work and, transcript H, identified here was not detected in their previous work [15]. These differences could be the result of distinct techniques used in our work (nested PCR) and in the previous work (cDNA cloning and sequencing). Alternatively, genetic differences in HCMV AD169 and TB40E (laboratory and clinical strains) may lead to differential alternative splicing patterns and production of different transcripts. Nevertheless, in MRC-5 cells transcripts A, B and E appear to be the most abundant, both in our work and in the previously published study [15].

Even though cIL-10 homologues have been identified in other members of the Herpesviridae family, HCMV UL111A is the only known viral cIL-10 homolog that expresses different IL-10 proteins by alternative splicing [44]. It is likely that HCMV evolved to regulate expression of these transcripts and proteins, which have different functions (i.e A and B), that counteract the immune system, in distinct phases of infection and in cells with different permissiveness for the virus. In this regard we decided to verify their expression in a GBM cell line. Previous studies have shown that established GBM cells, with different degrees of

differentiation, exhibit different levels of permissiveness to HCMV infection and replication [45–51]. In these cells, the kinetics of protein expression is delayed, and low numbers of progeny viruses are produced [49]. In addition, HCMV gene products and proteins have been detected in GBM tumors and various studies indicate that many viral genes products can be implicated in tumor malignity, including HCMV IL-10 [52].

Interestingly, in U251 GBM cells, we did not identify transcript A at any time point of infection. Transcript B is predominantly expressed at 24, 48 and 72, but not at 96 h. At 72 h, we observed transcripts F and H, which were also detected at 72 h in MRC-5 cells. We believe that further studies will be important to verify the presence, as well as the expression levels and kinetics, of the various viral IL-10 transcripts and proteins in cells with different degrees of permissiveness, including different cell types in which the virus establishes latency.

Cellular IL-10 signalling requires binding and formation of a cell surface complex formed by IL-10R1 and IL-10R2 chains. Initially, cIL-10 binds via a high affinity interaction with IL-10R1 that leads to exposure of sites for IL-10R2 binding on the complex surface. These interactions trigger the induction of STAT3 signalling via the phosphorylation of the cytoplasmic tails of IL-10R1 and IL-10R2 by JAK1 and Tyk2, respectively [53]. Regarding the HCMV IL-10 proteins, cmvIL-10 encoded by transcript A is the characterized the best characterized [54], and the only isoform with a reported crystal structure (Jones et al, 2002). cmvIL-10 binds to the receptor (IL-10R1) with the same affinity as cIL-10, inducing signalling that closely resembles that of cIL-10, despite low amino acid sequence identity between the two homologs (27%). According to Jones et al (2002), cmvIL-10 uses essentially the same structural epitope as cIL-10, comprised of helix A, the AB loop, and helix F, to contact IL-10R1 [8]. LAcmvIL-10 (isoform B) does not have all the properties of cmvIL-10 [14], and the other isoforms (C, D, E, F and G) have been poorly studied. However, Lin et al (2008) reported the expression of isoforms C to G at the protein level during HCMV infection and showed that they are unable to induce STAT3 phosphorylation in THP-1 cells [15].

The differences in protein signalling may be a consequence in the structure of the isoforms and their specific interaction with the receptor. To gain insights into the different proteins encoded by the UL111A gene and how they may interact with the IL-10R1 subunit of cIL-10 receptor, the three-dimensional structures of the isoforms, encoded by transcripts identified in our work, were modelled and compared to cIL-10 and cmvIL-10, pointing to key structural differences between them (Fig 2).

As expected, our modelled isoform A exhibits a structure highly similar to cmv-IL-10 and shares the residues reported to be important for binding to IL-10R1 [8]. In addition, we detected other residues that may also contribute to receptor binding as listed in S2 Table. LAcmvIL-10 does not have the C-terminal helices E and F, which are present in cmvIL-10 and cIL-10 [8,55]. In spite of this, we show that in place of helix F, LAcmvIL-10 contains a unique C-terminal extra helix, named here helix H (Fig 2). Helix H is not present in the other isoforms, even though the newly described isoform H exhibits amino acid conservation in this region (Fig 2). LAcmvIL-10 does not trigger STAT3 phosphorylation and does not have most of the biological properties of cmvIL-10. Nevertheless, it can inhibit MHC I class I and II transcription, even in the presence of anti-IL-10 receptor antibodies [14]. However, to the best of our knowledge, the exact binding sites of the neutralizing antibodies used in the Jenkins study are unknown.

It was previously suggested that the restricted signalling abilities of LAcmvIL-10 are likely due to the lack of helices E and F, which are present in cIL-10 and cmvIL-10. Furthermore, it was postulated that LAcmvIL-10 engages the receptor in a different manner [14]. Our analysis suggests that LAcmvIL-10 may interact with IL-10R1, and further experimental studies are necessary to confirm this interaction and the involvement of the extra unique helix (H) in a

possible complex. It is also possible that, as suggested, LAcmvIL-10 utilizes a different receptor or uses a receptor-independent mechanism for downregulation of MHC class II [14]. However, our data suggest that more likely LAcmvIL-10 signals through the cIL-10 receptor, perhaps in a conformation that is not disrupted by the neutralizing antibodies used by Jenkins et al (2008). It will be important to analyse different downstream pathways that could be activated through the cIL-10 in response to LAcmvIL-10.

Regarding the other isoforms, E lacks helix C, however, as reported, cmvIL-10 does not have residues important for engagement of IL-10R1 in helix C. In the same manner, isoform F lacks helix E, which also does not have residues required for IL-10R1 binding [8]. It is worth to mention that the amino acid residues with the highest contribution to binding to the receptor are not only in one region (binding site), but can be located along its structure, a factor that possibly enables the coupling of the different isoforms to the receptor, as shown in S2 Table. Therefore, our analysis indicates that isoforms E and F should also be able to bind the receptor subunit.

At last, the newly described isoform H contains only helices A and D and, although it retains amino acid sequence conservation to helices C, E, F and H, these helices are not formed in its final structure. These observations indicate that isoform H is the least likely to bind to the receptor, or it binds to IL-10R1 with low affinity. According to our computer simulation results, it is possible for all complexes to be formed. In the complexes formed between isoforms E, F, H and IL-10R1, six regions (possible binding sites) were detected, containing amino acid residues with affinity for all the probe molecules tested.

By coupling BINANA and normal modes calculation, we observed that the complex formed by isoform F and IL-10R1 showed higher stiffness and a more favourable receptor docking conformation than the other isoforms, suggesting higher stability compared to the complexes formed by the receptor and isoforms B, E, A and H, in order of binding affinity, respectively. One of the factors that may contribute to the stability between the isoform F and IL-10R1 are the T-shaped molecular interactions, which were not detected in the other complexes. Therefore, isoform F contains amino acid residues that contribute to its enhanced binding to the receptor. Furthermore, the position of the binding site in isoform F is more open than in the other isoforms, and this could facilitate its binding to the receptor, as observed by the molecular dynamic's trajectory of the analysed complexes. One can imagine that the ability to bind with higher affinity to the receptor could result in enhanced signalling or in competitive inhibition of binding by other isoforms to IL-10R1.

Interestingly, isoform B (LAcmvIL-10) has higher affinity than A (cmvIL-10) to IL-10R1 and this fact is curious, since the latter has more biological properties. Isoform A has four possible binding regions, and the sites 1 and 2 are close enough to be considered a larger site, which can possibly confer better binding to the receptor and signalling induction.

In sum, the results obtained in this study demonstrate that there is no structural impediment for the formation of complexes between the HCMV IL-10 isoforms with the IL-10R1 receptor. However, each isoform shows distinct molecular interactions, different energy values (ΔG) in the coupling, and different amino acid residues involved, making them a very peculiar biological system. Our results encourage more studies to be carried out on the HCMV IL-10 isoforms, contributing to the elucidation of the underlying mechanisms.

## Supporting information

**S1 Fig. Ramachandran plots.** General case Ramachandran plot for the HCMV IL-10 isoforms B, E, F and H.
(TIFF)

**S2 Fig. Particularities of the complex formed between the Isoform A and IL-10R1.** (A) Binding sites detected in the isoform. Site 1: Pink; site 2: Green; site 3: Lilac; site 4: Brown. (B) Representation of the isoforms on the surface. (C) Bioactive conformation of the isoforms coupled to the receptor. (D) Vectors obtained by calculating the normal modes, where we observe regions with less movement (more rigid) (blue), regions composed of residues with high movement (pink and red), and regions with intermediate movement (white). (TIF)

**S3 Fig. Particularities of the complex formed between the Isoform B and IL-10R1.** (A) Binding sites detected in the isoform. Site 1: Pink; site 2: Green; site 3: Lilac; site 4: Brown; site 5: Olive green. (B) Representation of the isoforms on the surface. (C) Bioactive conformation of the isoforms coupled to the receptor. (D) Vectors obtained by calculating the normal modes, where we observe regions with less movement (more rigid) (blue), regions composed of residues with high movement (pink and red), and regions with intermediate movement (white). (TIF)

**S4 Fig. Particularities of the complex formed between the Isoform E and the receptor.** (A) Binding sites detected in the isoform. Site 1: Pink; site 2: Green; site 3: Lilac; site 4: Brown; site 5: Olive green; site 6: Grey. (B) Representation of the isoforms on the surface. (C) Bioactive conformation of the isoforms coupled to the receptor. (D) Vectors obtained by calculating the normal modes, where we observe regions with less movement (more rigid) (blue), regions composed of residues with high movement (pink and red), and regions with intermediate movement (white). (TIF)

**S5 Fig. Particularities of the complex formed between the Isoform H and the receptor.** (A) Binding sites detected in the isoform. Site 1: Pink; site 2: Green; site 3: Lilac; site 4: Brown. (B) Representation of the isoforms on the surface. (C) Bioactive conformation of the isoforms coupled to the receptor. (D) Vectors obtained by calculating the normal modes, where we observe regions with less movement (more rigid) (blue), regions composed of residues with high movement (pink and red), and regions with intermediate movement (white). (TIF)

**S6 Fig. Difference in root mean square fluctuation (RMSF) values between isoform F (complexed with IL-10R1 and Apo) and isoforms A, B, E, F and H (Apo).** (TIF)

**S1 Table.** (A) Description of amino acid residues detected in each binding site of the studied proteins. (B) Sizing of the binding sites found. (DOCX)

**S2 Table. Main molecular interactions detected by BINANA for each receptor-coupled isoform.** (DOCX)

**S1 Raw images.** (PDF)

## Acknowledgments

We thank Dr. Eugenia Costanzi-Strauss, University of São Paulo-Brazil, for the GBM cells.

## Author Contributions

**Conceptualization:** Michael Nevels, Ana Ligia Scott.

**Data curation:** Simone Queiroz Pantaleão, Lívia de Moraes Bomediano Camillo, Tainan Cerqueira Neves, Lucas Matheus Stangherlin, Eric Alisson Philot, Ana Ligia Scott.

**Formal analysis:** Simone Queiroz Pantaleão, Lívia de Moraes Bomediano Camillo, Tainan Cerqueira Neves, Isabela de Godoy Menezes, Lucas Matheus Stangherlin, Maria Cristina Carlan da Silva.

**Funding acquisition:** Tainan Cerqueira Neves, Maria Cristina Carlan da Silva.

**Investigation:** Emma Poole, Michael Nevels, Maria Cristina Carlan da Silva.

**Methodology:** Simone Queiroz Pantaleão, Lívia de Moraes Bomediano Camillo, Isabela de Godoy Menezes, Helena Beatriz de Carvalho Ruthner Batista, Emma Poole, Michael Nevels, Maria Cristina Carlan da Silva.

**Project administration:** Maria Cristina Carlan da Silva.

**Software:** Simone Queiroz Pantaleão, Lívia de Moraes Bomediano Camillo.

**Supervision:** Maria Cristina Carlan da Silva.

**Validation:** Helena Beatriz de Carvalho Ruthner Batista, Michael Nevels, Ana Ligia Scott, Maria Cristina Carlan da Silva.

**Writing – original draft:** Lívia de Moraes Bomediano Camillo, Tainan Cerqueira Neves, Emma Poole, Ana Ligia Scott, Maria Cristina Carlan da Silva.

**Writing – review & editing:** Maria Cristina Carlan da Silva.

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
