## [Decision Letter · Decision Letter 0]

18 Aug 2022

PONE-D-22-17017Molecular modelling of the HCMV IL-10 protein isoforms and analysis of their interaction with the human IL-10 receptorPLOS ONE

Dear Dr. Carlan Silva,

Thank you for submitting your manuscript to PLOS ONE. After careful consideration, we feel that it has merit but does not fully meet PLOS ONE’s publication criteria as it currently stands. Therefore, we invite you to submit a revised version of the manuscript that addresses the points raised during the review process.

Your manuscript has been reviewed by two experts whose comments are appended below.  Overall the reviewers were enthusiastic about the manuscript and request only minor textual changes.  Please address all the reviewers' comments in your rebuttal letter.

We look forward to receiving your revised manuscript.

Kind regards,

Juliet V Spencer, Ph.D.

Academic Editor

PLOS ONE

Journal Requirements:

“Funder 01: Grant number: 2020/08527-1

Fundação de Amparo a Pesquisa do Estado de São Paulo, Brazil (FAPESP) - https://fapesp.br/

Author: Maria Cristina Carlan da Silva - Maria Cristina Carlan Silva

Funder 02: Grant number: 2020/07767-9

Fundação de Amparo a Pesquisa do Estado de São Paulo, Brazil (FAPESP) - https://fapesp.br/

Author: Tainan Cerqueira Neves - Neves, Tainan C.

Funder 03: Grant number: 164052/2020-8

Conselho Nacional de Desenvolvimento Científico e Tecnológico (CNPq) - https://www.gov.br/cnpq/pt-br

Author: Simone Queiroz Pantaleão - Pantaleão, Simone Queiroz”

Reviewers' comments:

Reviewer's Responses to Questions

**Comments to the Author**

1. Is the manuscript technically sound, and do the data support the conclusions?

Reviewer #1: Yes

Reviewer #2: Yes

2. Has the statistical analysis been performed appropriately and rigorously? 

Reviewer #1: N/A

Reviewer #2: N/A

3. Have the authors made all data underlying the findings in their manuscript fully available?

Reviewer #1: Yes

Reviewer #2: Yes

4. Is the manuscript presented in an intelligible fashion and written in standard English?

Reviewer #1: Yes

Reviewer #2: Yes

5. Review Comments to the Author

Reviewer #1: The study by Silva et al used computational methodology to perform molecular modeling of different vIL-10 isoforms with the IL-10R1. This interesting study highlights some intriguing aspects of vIL-10 biology that provides possible insight into the functions of different vIL-10 isoforms.

- Text describing experiments shown in Figs 4&5 would be improved by more clearly describing the rationale behind focusing on isoform F.

- Increased stability of Isoform B with IL-10R1 versus Isoform A is very curious. The discussion of this finding by the authors is good, however I think more explanation as to why Isoform A could still be more biologically active would improve the manuscript. What evidence or precedent is there that suggests that Isoform A intersection of binding sites 1 and 2 would enhance biological activity?

- The manuscript could be improved overall by some editing, with paragraphs needing editing (e.g., removal/editing of very short paragraphs).

Minor comments

-

- Line 315 Fig.2. Structure and aminoacid – needs a space

- 321 – gray > grey

- 552 - Therefore, despite of exhibiting – should read ‘in spite of’

Reviewer #2: In “Molecular Modelling of the HCMV IL-10 protein isoforms and analysis of their interaction with the human IL-10 receptor” (PLoS ONE manuscript #: PONE-D-22-17017) Dr. Silva and colleagues identify a new spliced transcript variant of the CMV UL111a gene (cmvIL-10). This transcript was identified in MRC-5 cells and a human glioblastoma multiforme cell line infected with low-passage HCMV strain TB40E. The authors follow-up their finding using in silica experiments designed to predict protein tertiary structure and IL-10R1 binding, comparing the newly identified isoform H with the previously described cmvIL-10 isoforms. The strength of the report is in the very nice transcriptional and sequencing analysis followed up with nice molecular modelling. The impact of the study is somewhat limited due to the in silico approach used to assess potential cmvIL-10 isoform interaction with IL-10R1. Also, the authors suggest strain variation or PCR technique as possible explanations for the absence of cmvIL-10 isoform H in previous publications. It would be interesting to see the authors perform their transcriptional and sequencing analysis approach on cells infected with AD169 to directly address this issue, although I don't think it is a requirement for publication.

Overall, the experiments are well designed and executed, and the proper controls are used throughout. The conclusions are supported by the experimental results. The manuscript is very well written, and all methods are described clearly and concisely. The main figures and layout are very nice as well, with only the very minor issue described below. This study will be of significant interest to scientists focused on the natural history of CMV, CMV host-pathogen interactions, and more broadly to immunologists interested in the IL-10 signaling cascade. Nice job!

Minor issues:

1. The Y-axis label in Fig 5A is clipped

6. PLOS authors have the option to publish the peer review history of their article (what does this mean?). If published, this will include your full peer review and any attached files.

Reviewer #1: No

Reviewer #2: No

---

## [Author Response · Author response to Decision Letter 0]

5 Oct 2022

General corrections

-The manuscript was revised to match the PLOS ONE’s style requirements. Including the files names.

-We included, in the financial disclosure, the following sentence. “The founders had no role in study design, data collection and analysis, decision to publish, or preparation of the manuscript”.

-The Data availability statement has been modified. “The group chose not to use public repository. The data will be available upon request via direct contact with the corresponding author”.

-We added the sequence identifier numbers of genbank in the manuscript.

-The Orcid ID was associated with the corresponding author of the manuscript in the PLOS ONE system.

-The gel and blot original images are available in the supplementary materials as a PDF file.

-Legends of the supplementary materials are available in the final of the manuscript.

-The bibliography has been revised

Reviewer's Responses to Questions responses to the specific comments.

Reviewer #1: 

- Text describing experiments shown in Figs 4&5 would be improved by more clearly describing the rationale behind focusing on isoform F.

Response: We better described the focus in isoform F in the Results / Molecular docking section and Discussion. This isoform likely stablish the most stable complex having a consistent increasing in its RMSF number when in its complexed conformation. One can imagine that the ability to bind with higher affinity to the receptor could result in enhanced signalling or in competitive inhibition of binding by other isoforms to IL-10R1, therefore we focused on isoform F. 

- Increased stability of Isoform B with IL-10R1 versus Isoform A is very curious. The discussion of this finding by the authors is good, however I think more explanation as to why Isoform A could still be more biologically active would improve the manuscript. What evidence or precedent is there that suggests that Isoform A intersection of binding sites 1 and 2 would enhance biological activity? 

Response: As far as we know there is no precedent evidence suggesting that intersection of binding sites 1 and 2 enhances the biological activity of isoform A. According to our data we suggest that this may occur. We can observe that isoform sites 1 and 2 in isoform A are very close to each other, possibly forming a large site, favouring connections and improving the binding possibilities. We describe it better throughout the text, mainly in the results section. 

- The manuscript could be improved overall by some editing, with paragraphs needing editing (e.g., removal/editing of very short paragraphs). 

Response: We performed a removal of short paragraphs and textual corrections. 

-Minor comments 

- Line 315 Fig.2. Structure and aminoacid – needs a space - Corrected 

- 321 – gray > grey - Corrected 

- 552 - Therefore, despite of exhibiting – should read ‘in spite of’ - Corrected 

Reviewer #2: 

The impact of the study is somewhat limited due to the in-silico approach used to assess potential cmvIL-10 isoform interaction with IL-10R1. Also, the authors suggest strain variation or PCR technique as possible explanations for the absence of cmvIL-10 isoform H in previous publications. It would be interesting to see the authors perform their transcriptional and sequencing analysis approach on cells infected with AD169 to directly address this issue, although I don't think it is a requirement for publication.

-Minor issues: 

1. The Y-axis label in Fig 5A is clipped - Corrected

---

## [Editor Report · Decision Letter 1]

10 Oct 2022

PONE-D-22-17017R1Molecular modelling of the HCMV IL-10 protein isoforms and analysis of their interaction with the human IL-10 receptorPLOS ONE

Dear Dr. Carlan Silva,

Thank you for submitting your manuscript to PLOS ONE. After careful consideration, we feel that it has merit but does not fully meet PLOS ONE’s publication criteria as it currently stands. Therefore, we invite you to submit a revised version of the manuscript that addresses the points raised during the review process.

Thank you for your attention to most of the reviewer comments.  However, there are still multiple "paragraphs" in your discussion that consist of 1 or 2 sentences.  The reviewers asked you to correct this and it has not been done.  A single sentence is not a paragraph.  Please seek assistance with editing from a native English speaker and pay careful attention to paragraph structure.  

We look forward to receiving your revised manuscript.

Kind regards,

Juliet V Spencer, Ph.D.

Academic Editor

PLOS ONE
---

## [Author Response · Author response to Decision Letter 1]

2 Nov 2022

The response to specific reviewer and editor comments are available in the attach files too.

Reviewer #1: 

- Text describing experiments shown in Figs 4&5 would be improved by more clearly describing the rationale behind focusing on isoform F.

Response: We better described the focus in isoform F in the Results / Molecular docking section and Discussion. This isoform likely stablish the most stable complex having a consistent increasing in its RMSF number when in its complexed conformation. One can imagine that the ability to bind with higher affinity to the receptor could result in enhanced signalling or in competitive inhibition of binding by other isoforms to IL-10R1, therefore we focused on isoform F.

- Increased stability of Isoform B with IL-10R1 versus Isoform A is very curious. The discussion of this finding by the authors is good, however I think more explanation as to why Isoform A could still be more biologically active would improve the manuscript. What evidence or precedent is there that suggests that Isoform A intersection of binding sites 1 and 2 would enhance biological activity?

Response: As far as we know there is no precendent evidence suggesting that intersection of binding sites 1 and 2 enhances the biological activity of isoform A. According to our data we suggest that this may occur. We can observe that isoform sites 1 and 2 in isoform A are very close to each other, possibly forming a large site, favouring connections and improving the binding possibilities.

We describe it better throughout the text, mainly in the results section. 

- The manuscript could be improved overall by some editing, with paragraphs needing editing (e.g., removal/editing of very short paragraphs).

Response: We performed a removal of short paragraphs and textual corrections.

Minor comments

- Line 315 Fig.2. Structure and aminoacid – needs a space

Corrected

- 321 – gray > grey

Corrected

- 552 - Therefore, despite of exhibiting – should read ‘in spite of’

Corrected

Reviewer #2:

The impact of the study is somewhat limited due to the in-silico approach used to assess potential cmvIL-10 isoform interaction with IL-10R1. 

Also, the authors suggest strain variation or PCR technique as possible explanations for the absence of cmvIL-10 isoform H in previous publications. 

It would be interesting to see the authors perform their transcriptional and sequencing analysis approach on cells infected with AD169 to directly address this issue, although I don't think it is a requirement for publication.

Minor issues:

1. The Y-axis label in Fig 5A is clipped

Corrected 

General corrections:

-The manuscript was revised to match the PLOS ONE’s style requirements. Including the files names.

-We included, in the financial disclosure, the following sentence. “The founders had no role in study design, data collection and analysis, decision to publish, or preparation of the manuscript”. 

-The Data availability statement has been modified. The data is available in https://www.kaggle.com/datasets/tainanneves/data-availability-hcmv-il10-molecular-models

-We added the sequence identifier numbers of genbank in the manuscript.

-The Orcid ID was associated with the corresponding author of the manuscript in the PLOS ONE system.

-The gel and blot original images are available in the supplementary materials as a PDF file.

-Legends of the supplementary materials are available in the final of the manuscript.

-The bibliography has been revised

---

## [Editor Report · Decision Letter 2]

8 Nov 2022

Molecular modelling of the HCMV IL-10 protein isoforms and analysis of their interaction with the human IL-10 receptor

PONE-D-22-17017R2

Dear Dr. Carlan Silva,

We’re pleased to inform you that your manuscript has been judged scientifically suitable for publication and will be formally accepted for publication once it meets all outstanding technical requirements.

Kind regards,

Juliet V Spencer, Ph.D.

Academic Editor

PLOS ONE
---

## [Editor Report · Acceptance letter]

15 Nov 2022

PONE-D-22-17017R2 

Molecular modelling of the HCMV IL-10 protein isoforms and analysis of their interaction with the human IL-10 receptor 

Dear Dr. Carlan da Silva:

I'm pleased to inform you that your manuscript has been deemed suitable for publication in PLOS ONE. Congratulations! Your manuscript is now with our production department. 

Kind regards, 

on behalf of

Dr. Juliet V Spencer 

Academic Editor

PLOS ONE